# Human Memory Search as Initial-Visit Emitting Random Walk

**Kwang-Sung Jun**[*], **Xiaojin Zhu**[†], **Timothy Rogers**[‡]
[*]Wisconsin Institute for Discovery, [†]Department of Computer Sciences, [‡]Department of Psychology
University of Wisconsin-Madison
`kjun@discovery.wisc.edu, jerryzhu@cs.wisc.edu, ttrogers@wisc.edu`

**Zhuoran Yang**
Department of Mathematical Sciences
Tsinghua University
`yzr11@mails.tsinghua.edu.cn`

**Ming Yuan**
Department of Statistics
University of Wisconsin-Madison
`myuan@stat.wisc.edu`

## Abstract

Imagine a random walk that outputs a state only when visiting it for the first time. The observed output is therefore a repeat-censored version of the underlying walk, and consists of a permutation of the states or a prefix of it. We call this model initial-visit emitting random walk (INVITE). Prior work has shown that the random walks with such a repeat-censoring mechanism explain well human behavior in memory search tasks, which is of great interest in both the study of human cognition and various clinical applications. However, parameter estimation in IN-VITE is challenging, because naive likelihood computation by marginalizing over infinitely many hidden random walk trajectories is intractable. In this paper, we propose the first efficient maximum likelihood estimate (MLE) for INVITE by decomposing the censored output into a series of absorbing random walks. We also prove theoretical properties of the MLE including identifiability and consistency. We show that INVITE outperforms several existing methods on real-world human response data from memory search tasks.

## 1 Human Memory Search as a Random Walk

A key goal for cognitive science has been to understand the mental structures and processes that underlie human semantic memory search. Semantic fluency has provided the central paradigm for this work: given a category label as a cue (e.g. animals, vehicles, etc.) participants must generate as many example words as possible in 60 seconds without repetition. The task is useful because, while exceedingly easy to administer, it yields rich information about human semantic memory. Participants do not generate responses in random order but produce "bursts" of related items, beginning with the highly frequent and prototypical, then moving to subclusters of related items. This ordinal structure sheds light on associative structures in memory: retrieval of a given item promotes retrieval of a related item, and so on, so that the temporal proximity of items in generated lists reflects the degree to which the two items are related in memory [14, 5]. The task also places demands on other important cognitive contributors to memory search: for instance, participants must retain a mental trace of previously-generated items and use it to refrain from repetition, so that the task draws upon working memory and cognitive control in addition to semantic processes. For these reasons the task is a central tool in all commonly-used metrics for diagnosing cognitive dysfunction (see e.g. [6]). Performance is generally sensitive to a variety of neurological disorders [19], but different syndromes also give rise to different patterns of impairment, making it useful for diagnosis [17]. For these reasons the task has been widely employed both in basic science and applied health research.

Nevertheless, the representations and processes that support category fluency remain poorly understood. Beyond the general observation that responses tend to be clustered by semantic relatedness,

it is not clear what ordinal structure in produced responses reveals about the structure of human semantic memory, in either healthy or disordered populations. In the past few years researchers in cognitive science have begun to fill this gap by considering how search models from other domains of science might explain patterns of responses observed in fluency tasks [12, 13, 15]. We review related works in Section 4.

In the current work we build on these advances by considering, not how search might operate on a pre-specified semantic representation, but rather how the representation itself can be learned from data (i.e., human-produced semantic fluency lists) given a specified model of the list-generation process. Specifically, we model search as a random walk on a set of states (e.g. words) where the transition probability indicates the strength of association in memory, and with the further constraint that node labels are only generated when the node is first visited. Thus, repeated visits are censored in the output. We refer to this generative process as the *initial-visit emitting* (INVITE) random walk. The repeat-censoring mechanism of INVITE was first employed in Abbott et al. [1]. However, their work did not provide a tractable method to compute the likelihood nor to estimate the transition matrix from the fluency responses. The problem of estimating the underlying Markov chain from the lists so produced is nontrivial because once the first two items in a list have been produced there may exist infinitely many pathways that lead to production of the next item. For instance, consider the produced sequence "dog" $\rightarrow$ "cat" $\rightarrow$ "goat" where the underlying graph is fully connected. Suppose a random walk visits "dog" then "cat". The walk can then visit "dog" and "cat" arbitrarily many times before visiting "goat"; there exist infinitely many walks that outputs the given sequence. How can the transition probabilities of the underlying random walk be learned?

A solution to this problem would represent a significant advance from prior works that estimate parameters from a separate source such as a standard text corpus [13]. First, one reason for verbal fluency's enduring appeal has been that the task appears to reveal important semantic structure that may not be discoverable by other means. It is not clear that methods for estimating semantic structure based on another corpus do a very good job at modelling the structure of human semantic representations generally [10], or that they would reveal the same structures that govern behavior specifically in this widely-used fluency task. Second, the representational structures employed can vary depending upon the fluency category. For instance, the probability of producing "chicken" after "goat" will differ depending on whether the task involves listing "animals", "mammals", or "farm animals". Simply estimating a single structure from the same corpus will not capture these task-based effects. Third, special populations, including neurological patients and developing children, may generate lists from quite different underlying mental representations, which cannot be independently estimated from a standard corpus.

In this work, we make two important contributions on the INVITE random walk. First, we propose a tractable way to compute the INVITE likelihood. Our key insight in computing the likelihood is to turn INVITE into a series of absorbing random walks. This formulation allows us to leverage the fundamental matrix [7] and compute the likelihood in polynomial time. Second, we show that the MLE of INVITE is consistent, which is non-trivial given that the convergence of the log likelihood function is not uniform. We formally define INVITE and present the two main contributions as well as an efficient optimization method to estimate the parameters in Section 2. In Section 3, we apply INVITE to both toy data and real-world fluency data. On toy data our experiments empirically confirm the consistency result. On actual human responses from verbal fluency INVITE outperforms off-the-shelf baselines. The results suggest that INVITE may provide a useful tool for investigating human cognitive functions.

## 2 The INVITE Random Walk

INVITE is a probabilistic model with the following generative story. Consider a random walk on a set of $n$ states $S$ with an initial distribution $\boldsymbol{\pi} > 0$ (entry-wise) and an arbitrary transition matrix $\mathbf{P}$ where $P_{ij}$ is the probability of jumping from state $i$ to $j$. A surfer starts from a random initial state drawn from $\boldsymbol{\pi}$. She outputs a state if it is the first time she visits that state. Upon arriving at an already visited state, however, she does not output the state. The random walk continues indefinitely. Therefore, the output consists of states in the order of their first-visit; the underlying entire walk trajectory is hidden. We further assume that the time step of each output is unobserved. For example, consider the random walk over four states in Figure 1(a). If the underlying random walk takes the trajectory $(1, 2, 1, 3, 1, 2, 1, 4, 1, \ldots)$, the observation is $(1, 2, 3, 4)$.

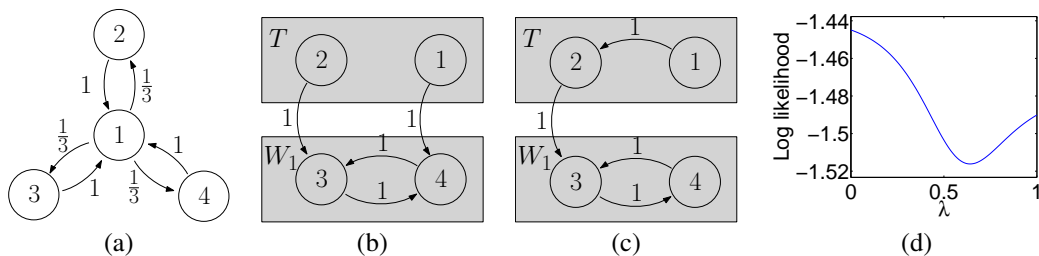

Figure 1: (a-c) Example Markov chains (d) Example nonconvexity of the INVITE log likelihood

We say that the observation produced by INVITE is a *censored* list since non-initial visits are censored. It is easy to see that a censored list is a permutation of the $n$ states or a prefix thereof (more on this later). We denote a censored list by $\mathbf{a} = (a_1, a_2, \ldots, a_M)$ where $M \leq n$. A censored list is not Markovian since the probability of a transition in censored list depends on the whole history rather than just the current state. It is worth noting that INVITE is distinct from Broder's algorithm for generating random spanning trees [4], or the self-avoiding random walk [9], or cascade models of infection. We discuss the technical difference to related works in Section 4.

We characterize the type of output INVITE is capable of producing, given that the underlying uncensored random walk continues indefinitely. A state $s$ is said to be *transient* if a random walk starting from $s$ has nonzero probability of not returning to itself in finite time and *recurrent* if such probability is zero. A set of states $A$ is *closed* if a walk cannot exit $A$; i.e., if $i \in A$ and $j \notin A$, then a random walk from $i$ cannot reach $j$. A set of states $B$ is *irreducible* if there exists a path between every pair of states in $B$; i.e., if $i, j \in B$, then a random walk from $i$ can reach $j$. Define $[M] = \{1, 2, \ldots, M\}$. We use $a_{1:M}$ as a shorthand for $a_1, \ldots, a_M$. Theorem 1 states that a finite state Markov chain can be uniquely decomposed into disjoint sets, and Theorem 2 states what a censored list should look like. All proofs are in the supplementary material.

**Theorem 1.** *[8] If the state space $S$ is finite, then $S$ can be written as a disjoint union $T \cup W_1 \cup \ldots \cup W_K$, where $T$ is a set of transient states that is possibly empty and each $W_k$, $k \in [K]$, is a nonempty closed irreducible set of recurrent states.*

**Theorem 2.** *Consider a Markov chain $\mathbf{P}$ with the decomposition $S = T \cup W_1 \cup \ldots \cup W_K$ as in Theorem 1. A censored list $\mathbf{a} = (a_{1:M})$ generated by INVITE on $\mathbf{P}$ has zero or more transient states, followed by all states in one and only one closed irreducible set. That is, $\exists \ell \in [M]$ s.t. $\{a_{1:\ell-1}\} \subseteq T$ and $\{a_{\ell:M}\} = W_k$ for some $k \in [K]$.*

As an example, when the graph is fully connected INVITE is capable of producing all $n!$ permutations of the $n$ states as the censored lists. As another example, in Figure 1 (b) and (c), both chains have two transient states $T = \{1, 2\}$ and two recurrent states $W_1 = \{3, 4\}$. (b) has no path that visits both 1 and 2, and thus every censored list must be a prefix of a permutation. However, (c) has a path that visits both 1 and 2, thus can generate (1,2,3,4), a full permutation.

In general, each INVITE run generates a permutation of $n$ states, or a prefix of a permutation. Let $\mathrm{Sym}(n)$ be the symmetric group on $[n]$. Then, the data space $\mathcal{D}$ of censored lists is $\mathcal{D} \equiv \{(a_{1:k}) \mid \mathbf{a} \in \mathrm{Sym}(n), k \in [n]\}$.

## 2.1 Computing the INVITE likelihood

Learning and inference under the INVITE model is challenging due to its likelihood function. A naive method to compute the probability of a censored list $\mathbf{a}$ given $\boldsymbol{\pi}$ and $\mathbf{P}$ is to sum over all uncensored random walk trajectories $\mathbf{x}$ which produces $\mathbf{a}$: $\mathbb{P}(\mathbf{a}; \boldsymbol{\pi}, \mathbf{P}) = \sum_{\mathbf{x} \text{ produces } \mathbf{a}} \mathbb{P}(\mathbf{x}; \boldsymbol{\pi}, \mathbf{P})$. This naive computation is intractable since the summation can be over an infinite number of trajectories $\mathbf{x}$'s that might have produced the censored list $\mathbf{a}$. For example, consider the censored list $\mathbf{a} = (1, 2, 3, 4)$ generated from Figure 1(a). There are infinite uncensored trajectories to produce $\mathbf{a}$ by visiting states 1 and 2 arbitrarily many times before visiting state 3, and later state 4.

The likelihood of $\boldsymbol{\pi}$ and $\mathbf{P}$ on a censored list $\mathbf{a}$ is

$$\mathbb{P}(\mathbf{a}; \boldsymbol{\pi}, \mathbf{P}) = \begin{cases} \pi_{a_1} \prod_{k=1}^{M-1} \mathbb{P}(a_{k+1} \mid a_{1:k}; \mathbf{P}) & \text{if } \mathbf{a} \text{ cannot be extended} \\ 0 & \text{otherwise.} \end{cases} \tag{1}$$

Note we assign zero probability to a censored list that is not completed yet, since the underlying random walk must run forever. We say a censored list $\mathbf{a}$ is *valid* (*invalid*) under $\boldsymbol{\pi}$ and $\mathbf{P}$ if $\mathbb{P}(\mathbf{a}; \boldsymbol{\pi}, \mathbf{P}) > 0 \ (= 0)$.

We first review the *fundamental matrix* in the absorbing random walk. A state that transits to itself with probability 1 is called an *absorbing state*. Given a Markov chain $\mathbf{P}$ with absorbing states, we can rearrange the states into $\mathbf{P}' = \begin{pmatrix} \mathbf{Q} & \mathbf{R} \\ \mathbf{0} & \mathbf{I} \end{pmatrix}$, where $\mathbf{Q}$ is the transition between the nonabsorbing states, $\mathbf{R}$ is the transition from the nonabsorbing states to absorbing states, and the rest trivially represent the absorbing states. Theorem 3 presents the *fundamental matrix*, the essential tool for the tractable computation of the INVITE likelihood.

**Theorem 3.** *[7] The fundamental matrix of the Markov chain $\mathbf{P}'$ is $\mathbf{N} = (\mathbf{I} - \mathbf{Q})^{-1}$. $N_{ij}$ is the expected number of times that a chain visits state $j$ before absorption when starting from $i$. Furthermore, define $\mathbf{B} = (\mathbf{I} - \mathbf{Q})^{-1}\mathbf{R}$. Then, $B_{ik}$ is the probability of a chain starting from $i$ being absorbed by $k$. In other words, $B_{i\cdot}$ is the absorption distribution of a chain starting from $i$.*

As a tractable way to compute the likelihood, we propose a novel formulation that turns an INVITE random walk into a series of absorbing random walks. Although INVITE itself is not an absorbing random walk, each segment that produces the next item in the censored list can be modeled as one. That is, for each $k = 1 \ldots M - 1$ consider the segment of the uncensored random walk starting from the previous output $a_k$ until the next output $a_{k+1}$. For this segment, we construct an absorbing random walk by keeping $a_{1:k}$ nonabsorbing and turning the rest into the absorbing states. A random walk starting from $a_k$ is eventually absorbed by a state in $S \setminus \{a_{1:k}\}$. The probability of being absorbed by $a_{k+1}$ is exactly the probability of outputting $a_{k+1}$ after outputting $a_{1:k}$ in INVITE. Formally, we construct an absorbing random walk $\mathbf{P}^{(k)}$:

$$\mathbf{P}^{(k)} = \begin{pmatrix} \mathbf{Q}^{(k)} & \mathbf{R}^{(k)} \\ \mathbf{0} & \mathbf{I} \end{pmatrix}, \tag{2}$$

where the states are ordered as $a_{1:M}$. Corollary 1 summarizes our computation of the INVITE likelihood.

**Corollary 1.** *The $k$-th step INVITE likelihood for $k \in [M-1]$ is*

$$\mathbb{P}(a_{k+1} \mid a_{1:k}, \mathbf{P}) = \begin{cases} [(\mathbf{I} - \mathbf{Q}^{(k)})^{-1}\mathbf{R}^{(k)}]_{k1} & \text{if } (\mathbf{I} - \mathbf{Q}^{(k)})^{-1} \text{ exists} \\ 0 & \text{otherwise} \end{cases} \tag{3}$$

Suppose we observe $m$ independent realizations of INVITE: $D_m = \left\{ \left(a_1^{(1)}, ..., a_{M_1}^{(1)}\right), ..., \left(a_1^{(m)}, ..., a_{M_m}^{(m)}\right) \right\}$, where $M_i$ is the length of the $i$-th censored list. Then, the INVITE log likelihood is $\ell(\boldsymbol{\pi}, \mathbf{P}; D_m) = \sum_{i=1}^{m} \log \mathbb{P}(\mathbf{a}^{(i)}; \boldsymbol{\pi}, \mathbf{P})$.

## 2.2 Consistency of the MLE

Identifiability is an essential property for a model to be consistent. Theorem 4 shows that allowing self-transitions in $\mathbf{P}$ cause INVITE to be unidentifiable. Then, Theorem 5 presents a remedy. The proof for both theorems are presented in our supplementary material. Let $\text{diag}(\mathbf{q})$ be a diagonal matrix whose $i$-th diagonal entry is $q_i$.

**Theorem 4.** *Let $\mathbf{P}$ be an $n \times n$ transition matrix without any self-transition ($P_{ii} = 0, \forall i$), and $\mathbf{q} \in [0, 1)^n$. Define $\mathbf{P}' = \text{diag}(\mathbf{q}) + (\mathbf{I} - \text{diag}(\mathbf{q}))\mathbf{P}$, a scaled transition matrix with self-transition probabilities $\mathbf{q}$. Then, $\mathbb{P}(\mathbf{a}; \boldsymbol{\pi}, \mathbf{P}) = \mathbb{P}(\mathbf{a}; \boldsymbol{\pi}, \mathbf{P}')$, for every censored list $\mathbf{a}$.*

For example, consider a censored list $\mathbf{a} = (1, j)$ where $j \neq 1$. Using the fundamental matrix, $\mathbb{P}(a_2 \mid a_1; \mathbf{P}) = (1 - P_{11})^{-1} P_{1j} = (\sum_{j' \neq 1} P_{1j'})^{-1} P_{1j} = (\sum_{j' \neq 1} cP_{1j'})^{-1} cP_{1j}, \forall c$. This implies that multiplying a constant $c$ to $P_{1j}$ for all $j \neq 1$ and renormalizing the first row $\mathbf{P}_{1\cdot}$ to sum to 1 does not change the likelihood.

**Theorem 5.** *Assume the initial distribution $\boldsymbol{\pi} > 0$ elementwise. In the space of transition matrices $\mathbf{P}$ without self-transitions, INVITE is identifiable.*

Let $\Delta^{n-1} = \{\mathbf{p} \in \mathbb{R}^n \mid p_i \geq 0, \forall i, \sum_i p_i = 1\}$ be the probability simplex. For brevity, we pack the parameters of INVITE into one vector $\boldsymbol{\theta}$ as follows: $\boldsymbol{\theta} \in \Theta = \{(\boldsymbol{\pi}^\top, \mathbf{P}_{1\cdot}, \ldots, \mathbf{P}_{n\cdot})^\top \mid \boldsymbol{\pi}, \mathbf{P}_{i\cdot} \in \Delta^{n-1}, P_{ii} = 0, \forall i\}$. Let $\boldsymbol{\theta}^* = (\boldsymbol{\pi}^{*\top}, \mathbf{P}_{1\cdot}^*, \ldots, \mathbf{P}_{n\cdot}^*)^\top \in \Theta$ be the true model. Given a set of $m$ censored lists $D_m$ generated from $\boldsymbol{\theta}^*$, the average log likelihood function and its pointwise limit are

$$\widehat{\mathcal{Q}}_m(\boldsymbol{\theta}) = \frac{1}{m} \sum_{i=1}^{m} \log \mathbb{P}(\mathbf{a}^{(i)}); \boldsymbol{\theta}) \qquad \text{and} \qquad \mathcal{Q}^*(\boldsymbol{\theta}) = \sum_{\mathbf{a} \in \mathcal{D}} \mathbb{P}(\mathbf{a}; \boldsymbol{\theta}^*) \log \mathbb{P}(\mathbf{a}; \boldsymbol{\theta}). \qquad (4)$$

For brevity, we assume that the true model $\boldsymbol{\theta}^*$ is strongly connected; the analysis can be easily extended to remove it. Under the Assumption A1, Theorem 6 states the consistency result.

**Assumption A1.** *Let* $\boldsymbol{\theta}^* = (\boldsymbol{\pi}^{*\top}, \mathbf{P}_1^*, \dots, \mathbf{P}_n^*)^\top \in \boldsymbol{\Theta}$ *be the true model.* $\boldsymbol{\pi}^*$ *has no zero entries. Furthermore,* $\mathbf{P}^*$ *is strongly connected.*

**Theorem 6.** *Assume A1. The MLE of INVITE* $\widehat{\boldsymbol{\theta}}_m \equiv \max_{\boldsymbol{\theta} \in \boldsymbol{\Theta}} \widehat{\mathcal{Q}}_m(\boldsymbol{\theta})$ *is consistent.*

We provide a sketch here. The proof relies on Lemma 6 and Lemma 2 that are presented in our supplementary material. Since $\boldsymbol{\Theta}$ is compact, the sequence $\{\widehat{\boldsymbol{\theta}}_m\}$ has a convergent subsequence $\{\widehat{\boldsymbol{\theta}}_{m_j}\}$. Let $\boldsymbol{\theta}' = \lim_{j \to \infty} \widehat{\boldsymbol{\theta}}_{m_j}$. Since $\widehat{\mathcal{Q}}_{m_j}(\boldsymbol{\theta}^*) \leq \widehat{\mathcal{Q}}_{m_j}(\widehat{\boldsymbol{\theta}}_{m_j})$,

$$\mathcal{Q}^*(\boldsymbol{\theta}^*) = \lim_{j \to \infty} \widehat{\mathcal{Q}}_{m_j}(\boldsymbol{\theta}^*) \leq \lim_{j \to \infty} \widehat{\mathcal{Q}}_{m_j}(\widehat{\boldsymbol{\theta}}_{m_j}) = \mathcal{Q}^*(\boldsymbol{\theta}'),$$

where the last equality is due to Lemma 6. By Lemma 2, $\boldsymbol{\theta}^*$ is the unique maximizer of $\mathcal{Q}^*$, which implies $\boldsymbol{\theta}' = \boldsymbol{\theta}^*$. Note that the subsequence was chosen arbitrarily. Since every convergent subsequence converges to $\boldsymbol{\theta}^*$, $\widehat{\boldsymbol{\theta}}_m$ converges to $\boldsymbol{\theta}^*$.

## 2.3 Parameter Estimation via Regularized Maximum Likelihood

We present a regularized MLE (RegMLE) of INVITE. We first extend the censored lists that we consider. Now we allow the underlying walk to terminate after finite steps because in real-world applications the observed censored lists are often truncated. That is, the underlying random walk can be stopped before exhausting every state the walk could visit. For example, in verbal fluency, participants have limited time to produce a list. Consequently, we use the *prefix likelihood*

$$\mathcal{L}(\mathbf{a}; \boldsymbol{\pi}, \mathbf{P}) = \pi_{a_1} \prod_{k=1}^{M-1} \mathbb{P}(a_{k+1} \mid a_{1:k}; \mathbf{P}). \qquad (5)$$

We find the RegMLE by maximizing the prefix log likelihood plus a regularization term on $\boldsymbol{\pi}, \mathbf{P}$. Note that, $\boldsymbol{\pi}$ and $\mathbf{P}$ can be separately optimized. For $\boldsymbol{\pi}$, we place a Dirichlet prior and find the maximum a posteriori (MAP) estimator $\widehat{\boldsymbol{\pi}}$ by $\widehat{\pi}_j \propto \sum_{i=1}^{m} \mathbb{1}_{a_1^{(i)} = j} + C_{\boldsymbol{\pi}}, \forall j$.

Directly computing the RegMLE of $\mathbf{P}$ requires solving a constrained optimization problem, because the transition matrix $\mathbf{P}$ must be row stochastic. We re-parametrize $\mathbf{P}$ which leads to a more convenient unconstrained optimization problem. Let $\boldsymbol{\beta} \in \mathbb{R}^{n \times n}$. We exponentiate $\boldsymbol{\beta}$ and row-normalize it to derive $\mathbf{P}$: $P_{ij} = e^{\beta_{ij}} / \sum_{j'=1}^{n} e^{\beta_{ij'}}, \forall i, j$. We fix the diagonal entries of $\boldsymbol{\beta}$ to $-\infty$ to disallow self-transitions. We place squared $\ell_2$ norm regularizer on $\boldsymbol{\beta}$ to prevent overfitting. The unconstrained optimization problem is:

$$\min_{\boldsymbol{\beta}} \quad -\sum_{i=1}^{m} \sum_{k=1}^{M_i-1} \log \mathbb{P}(a_{k+1}^{(i)} \mid a_{1:k}^{(i)}; \boldsymbol{\beta}) + \frac{1}{2} C_{\boldsymbol{\beta}} \sum_{i \neq j} \beta_{ij}^2 \quad , \qquad (6)$$

where $C_{\boldsymbol{\beta}} > 0$ is a regularization parameter. We provide the derivative of the prefix log likelihood w.r.t. $\boldsymbol{\beta}$ in our supplementary material. We point out that the objective function of (6) is not convex in $\boldsymbol{\beta}$ in general. Let $n = 5$ and suppose we observe two censored lists $(5, 4, 3, 1, 2)$ and $(3, 4, 5, 1, 2)$. We found with random starts two different local optima $\boldsymbol{\beta}^{(1)}$ and $\boldsymbol{\beta}^{(2)}$ of (6). We plot the prefix log likelihood of $(1 - \lambda)\boldsymbol{\beta}^{(1)} + \lambda\boldsymbol{\beta}^{(2)}$, where $\lambda \in [0, 1]$ in Figure 1(d). Nonconvexity of this 1D slice implies nonconvexity of the prefix log likelihood surface in general.

**Efficient Optimization using Averaged Stochastic Gradient Descent** Given a censored list $\mathbf{a}$ of length $M$, computing the derivative of $\mathbb{P}(a_{k+1} \mid a_{1:k})$ w.r.t. $\boldsymbol{\beta}$ takes $O(k^3)$ time for matrix inversion. There are $n^2$ entries in $\boldsymbol{\beta}$, so the time complexity per item is $O(k^3 + n^2)$. This computation needs to be done for $k = 1, \dots, (M-1)$ in a list and for $m$ censored lists, which makes the overall time complexity $O(mM(M^3 + n^2))$. In the worst case, $M$ is as large as $n$, which makes it $O(mn^4)$. Even the state-of-the-art batch optimization method such as LBFGS takes a very long time to find the solution for a moderate problem size such as $n \approx 500$. For a faster computation of the RegMLE (6), we turn to averaged stochastic gradient descent (ASGD) [20, 18]. ASGD processes the lists sequentially by updating the parameters after every list. The per-round objective function for $\boldsymbol{\beta}$ on the $i$-th list is

$$f(\mathbf{a}^{(i)}; \boldsymbol{\beta}) \equiv -\sum_{k=1}^{M_i-1} \log \mathbb{P}(a_{k+1}^{(i)} \mid a_{1:k}^{(i)}; \boldsymbol{\beta}) + \frac{C_{\boldsymbol{\beta}}}{2m} \sum_{i \neq j} \beta_{ij}^2.$$

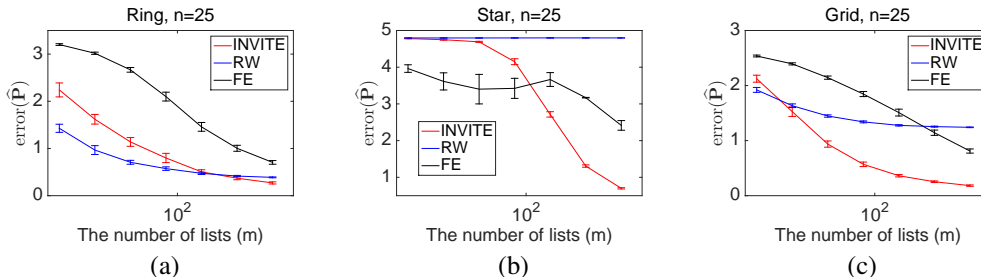

Figure 2: Toy experiment results where the error is measured with the Frobenius norm.

We randomly initialize $\boldsymbol{\beta}_0$. At round $t$, we update the solution $\boldsymbol{\beta}_t$ with $\boldsymbol{\beta}_t \leftarrow \boldsymbol{\beta}_{t-1} - \eta_t \nabla f(a^{(i)}; \boldsymbol{\beta})$ and the average estimate $\overline{\boldsymbol{\beta}}_t$ with $\overline{\boldsymbol{\beta}}_t \leftarrow \frac{t-1}{t}\overline{\boldsymbol{\beta}}_{t-1} + \frac{1}{t}\boldsymbol{\beta}_t$. Let $\eta_t = \gamma_0(1 + \gamma_0 a t)^{-c}$. We use $a = C_{\boldsymbol{\beta}}/m$ and $c = 3/4$ following [3] and pick $\gamma_0$ by running the algorithm on a small subsample of the train set. We run ASGD for a fixed number of epochs and take the final $\overline{\boldsymbol{\beta}}_t$ as the solution.

## 3 Experiments

We compare INVITE against two popular estimators of $\mathbf{P}$: naive random walk (RW) and First-Edge (FE). RW is the regularized MLE of the naive random walk, pretending the censored lists are the underlying uncensored walk trajectory: $\widehat{P}_{rc}^{(RW)} \propto \left( \sum_{i=1}^{m} \sum_{j=1}^{M_i-1} \mathbb{1}_{(a_j^{(i)}=r) \wedge (a_{j+1}^{(i)}=c)} \right) + C_{RW}$. Though simple and popular, RW is a biased estimator due to the model mismatch. FE was proposed in [2] for graph structure recovery in cascade model. FE uses only the first two items in each censored list: $\widehat{P}_{rc}^{(FE)} \propto \left( \sum_{i=1}^{m} \mathbb{1}_{(a_1^{(i)}=r) \wedge (a_2^{(i)}=c)} \right) + C_{FE}$. Because the first transition in a censored list is always the same as the first transition in its underlying trajectory, FE is a consistent estimator of $\mathbf{P}$ (assuming $\boldsymbol{\pi}$ has no zero entries). In fact, FE is equivalent to the RegMLE of the length two prefix likelihood of the INVITE model. However, we expect FE to waste information since it discards the rest of the censored lists. Furthermore, FE cannot estimate the transition probabilities from an item that does not appear as the first item in the lists, which is common in real-world data.

### 3.1 Toy Experiments

Here we compare the three estimators INVITE, RW, and FE on toy datasets, where the observations are indeed generated by an initial-visit emitting random walk. We construct three undirected, unweighted graphs of $n = 25$ nodes each: (i) **Ring**, a ring graph, (ii) **Star**, $n-1$ nodes each connected to a "hub" node, and (iii) **Grid**, a 2-dimensional $\sqrt{n} \times \sqrt{n}$ lattice.

The initial distribution $\boldsymbol{\pi}^*$ is uniform, and the transition matrix $\mathbf{P}^*$ at each node has an equal transition probability to its neighbors. For each graph, we generate datasets with $m \in \{10, 20, 40, 80, 160, 320, 640\}$ censored lists. Each censored list has length $n$. We note that, in the star graph a censored list contains many apparent transitions between leaf nodes, although such transitions are not allowed in its underlying uncensored random walk. This will mislead RW. This effect is less severe in the grid graph and the ring graph.

For each estimator, we perform 5-fold cross validation (CV) for finding the best smoothing parameters $C_{\boldsymbol{\beta}}, C_{RW}, C_{FE}$ on the grid $10^{-2}, 10^{-1.5}, \dots, 10^1$, respectively, with which we compute each estimator. Then, we evaluate the three estimators using the Frobenius norm between $\widehat{\mathbf{P}}$ and the true transition matrix $\mathbf{P}^*$: $\text{error}(\widehat{\mathbf{P}}) = \sqrt{\sum_{i,j}(\widehat{P}_{ij} - P_{ij}^*)^2}$. Note the error must approach 0 as $m$ increases for consistent estimators. We repeat the same experiment 20 times where each time we draw a new set of censored lists.

Figure 2 shows how $\text{error}(\widehat{\mathbf{P}})$ changes as the number of censored lists $m$ increases. The error bars are 95% confidence bounds. We make three observations: (1) *INVITE tends towards 0 error*. This is expected given the consistency of INVITE in Theorem 6. (2) *RW is biased*. In all three plots, RW tends towards some positive number, unlike INVITE and FE. This is because RW has the wrong model on the censored lists. (3) *INVITE outperforms FE*. On the ring and grid graphs INVITE dominates FE for every training set size. On the star graph FE is better than INVITE with a small $m$, but INVITE eventually achieves lower error. This reflects the fact that, although FE is unbiased, it discards most of the censored lists and therefore has higher variance compared to INVITE.

| | Animal | Food |
|---|---|---|
| $n$ | 274 | 452 |
| $m$ | 4710 | 4622 |
| **Length** Min. | 2 | 1 |
| Max. | 36 | 47 |
| Mean | 18.72 | 20.73 |
| Median | 19 | 21 |

Table 1: Statistics of the verbal fluency data.

| | Model | Test set mean neg. loglik. |
|---|---|---|
| | INVITE | **60.18 ($\pm$1.75)** |
| Animal | RW | 69.16 ($\pm$2.00) |
| | FE | 72.12 ($\pm$2.17) |
| | INVITE | **83.62 ($\pm$2.32)** |
| Food | RW | 94.54 ($\pm$2.75) |
| | FE | 100.27 ($\pm$2.96) |

Table 2: Verbal fluency test set log likelihood.

## 3.2 Verbal Fluency

We now turn to the real-world fluency data where we compare INVITE with the baseline models. Since we do not have the ground truth parameter $\pi$ and $\mathbf{P}$, we compare test set log likelihood of various models. Confirming the empirical performance of INVITE sheds light on using it for practical applications such as the dignosis and classification of the brain-damaged patient.

**Data** The data used to assess human memory search consists of two verbal fluency datasets from the Wisconsin Longitudinal Survey (WLS). The WLS is a longitudinal assessment of many sociodemographic and health factors that has been administered to a large cohort of Wisconsin residents every five years since the 1950s. Verbal fluency for two semantic categories, animals and foods, was administered in the last two testing rounds (2005 and 2010), yielding a total of 4714 lists for animals and 4624 lists for foods collected from a total of 5674 participants ranging in age from their early-60's to mid-70's. The raw lists included in the WLS were preprocessed by expanding abbreviations ("lab" → "labrador"), removing inflections ("cats" → "cat"), correcting spelling errors, and removing response errors like unintelligible items. Though instructed to not repeat, some human participants did occasionally produce repeated words. We removed the repetitions from the data, which consist of 4% of the word token responses. Finally, the data exhibits a Zipfian behavior with many idiosyncratic, low count words. We removed words appearing in less than 10 lists. In total, the process resulted in removing 5% of the total number of word token responses. The statistics of the data after preprocessing is summarized in Table 1.

**Procedure** We randomly subsample 10% of the lists as the test set, and use the rest as the training set. We perform 5-fold CV on the training set for each estimator to find the best smoothing parameter $C_{\boldsymbol{\beta}}, C_{RW}, C_{FE} \in \{10^1, 10^{.5}, 10^0, 10^{-.5}, 10^{-1}, 10^{-1.5}, 10^{-2}\}$ respectively, where the validation measure is the prefix log likelihood for INVITE and the standard random walk likelihood for RW. For the validation measure of FE we use the INVITE prefix log likelihood since FE is equivalent to the length two prefix likelihood of INVITE. Then, we train the final estimator on the whole training set using the fitted regularization parameter.

**Result** The experiment result is summarized in Table 2. For each estimator, we measure the average per-list negative prefix log likelihood on the test set for INVITE and FE, and the standard random walk per-list negative log likelihood for RW. The number in the parenthesis is the 95% confidence interval. Boldfaced numbers mean that the corresponding estimator is the best and the difference from the others is statistically significant under a two-tailed paired $t$-test at 95% significance level. In both animal and food verbal fluency tasks, the result indicates that human-generated fluency lists are better explained by INVITE than by either RW or FE. Furthermore, RW outperforms FE. We believe that FE performs poorly despite being consistent because the number of lists is too small (compared to the number of states) for FE to reach a good estimate.

## 4 Related Work

Though behavior in semantic fluency tasks has been studied for many years, few computationally explicit models of the task have been advanced. Influential models in the psychological literature, such as the widely-known "clustering and switching" model of Troyer et al. [21], have been articulated only verbally. Efforts to estimate the structure of semantic memory from fluency lists have mainly focused on decomposing the structure apparent in distance matrices that reflect the mean inter-item ordinal distances across many fluency lists [5]—but without an account of the processes that generate list structure it is not clear how the results of such studies are best interpreted. More recently, researchers in cognitive science have begun to focus on explicit model of the processes by which fluency lists are generated. In these works, the structure of semantic memory is first modelled either as a graph or as a continuous multidimensional space estimated from word co-occurrence statistics in large corpora of natural language. Researchers then assess whether structure in fluency data can be understood as resulting from a particular search process operating over the specified semantic

structure. Models explored in this vein include simple random walk over a semantic network, with repeated nodes omitted from the sequence produced [12], the PageRank algorithm employed for network search by Google [13], and foraging algorithms designed to explain the behavior of animals searching for food [15]. Each example reports aspects of human behavior that are well-explained by the respective search process, given accompanying assumptions about the nature of the underlying semantic structure. However, these works do not learn their model directly from the fluency lists, which is the key difference from our study.

Broder's algorithm **Generate** [4] for generating random spanning tree is similar to INVITE's generative process. Given an undirected graph, the algorithm runs a random walk and outputs each transition to an unvisited node. Upon transiting to an already visited node, however, it does not output the transition. The random walk stops after visiting every node in the graph. In the end, we observe an ordered list of transitions. For example, in Figure 1(a) if the random walk trajectory is (2,1,2,1,3,1,4), then the output is (2→1, 1→3, 1→4). Note that if we take the starting node of the first transition and the arriving nodes of each transition, then the output list reduces to a censored list generated from INVITE with the same underlying random walk. Despite the similarity, to the best of our knowledge, the censored list derived from the output of the algorithm Generate has not been studied, and there has been no parameter estimation task discussed in prior works.

Self-avoiding random walk, or non-self-intersecting random walk, performs random walk while avoiding already visited node [9]. For example, in Figure 1(a), if a self-avoiding random walk starts from state 2 then visits 1, then it can only visit states 3 or 4 since 2 is already visited. In not visiting the same node twice, self-avoiding walk is similar to INVITE. However, a key difference is that self-avoiding walk cannot produce a transition $i \rightarrow j$ if $P_{ij} = 0$. In contrast, INVITE can appear to have such "transitions" in the censored list. Such behavior is a core property that allows INVITE to *switch* clusters in modeling human memory search.

INVITE resembles cascade models in many aspects [16, 11]. In a cascade model, the information or disease spreads out from a seed node to the whole graph by infections that occur from an infected node to its neighbors. [11] formulates a graph learning problem where an observation is a list, or so-called trace, that contains infected nodes along with their infection time. Although not discussed in the present paper, it is trivial for INVITE to produce time stamps for each item in its censored list, too. However, there is a fundamental difference in how the infection occurs. A cascade model typically allows multiple infected nodes to infect their neighbors in parallel, so that infection can happen simultaneously in many parts of the graph. On the other hand, INVITE contains a *single surfer* that is responsible for all the infection via a random walk. Therefore, infection in INVITE is necessarily sequential. This results in INVITE exhibiting clustering behaviors in the censored lists, which is well-known in human memory search tasks [21].

## 5 Discussion

There are numerous directions to extend INVITE. First, more theoretical investigation is needed. For example, although we know the MLE of INVITE is consistent, the convergence rate is unknown. Second, one can improve the INVITE estimate when data is sparse by assuming certain cluster structures in the transition matrix $\mathbf{P}$, thereby reducing the degrees of freedom. For instance, it is known that verbal fluency tends to exhibit "runs" of semantically related words. One can assume a stochastic block model $\mathbf{P}$ with parameter sharing at the block level, where the blocks represent semantic clusters of words. One then estimates the block structure and the shared parameters at the same time. Third, INVITE can be extended to allow repetitions in a list. The basic idea is as follows. In the $k$-th segment we previously used an absorbing random walk to compute $\mathbb{P}(a_{k+1} \mid a_{1:k})$, where $a_{1:k}$ were the nonabsorbing states. For each nonabsorbing state $a_i$, add a "dongle twin" absorbing state $a'_i$ attached only to $a_i$. Allow a small transition probability from $a_i$ to $a'_i$. If the walk is absorbed by $a'_i$, we output $a_i$ in the censored list, which becomes a repeated item in the censored list. Note that the likelihood computation in this augmented model is still polynomial. Such a model with "reluctant repetitions" will be an interesting interpolation between "no repetitions" and "repetitions as in a standard random walk."

**Acknowledgments**

The authors are thankful to the anonymous reviewers for their comments. This work is supported in part by NSF grants IIS-0953219 and DGE-1545481, NIH Big Data to Knowledge 1U54AI117924-01, NSF Grant DMS-1265202, and NIH Grant 1U54AI117924-01.

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
