[Supplementary Material]

# Supplementary Material for "Human Memory Search as Initial-Visit Emitting Random Walk"

**Kwang-Sung Jun**[*], **Xiaojin Zhu**[†], **Timothy Rogers**[‡]
[*]Wisconsin Institute for Discovery, [†]Department of Computer Sciences, [‡]Department of Psychology
University of Wisconsin-Madison
kjun@discovery.wisc.edu, jerryzhu@cs.wisc.edu, ttrogers@wisc.edu

**Zhuoran Yang**
Department of Mathematical Sciences
Tsinghua University
yzr11@mails.tsinghua.edu.cn

**Ming Yuan**
Department of Statistics
University of Wisconsin-Madison
myuan@stat.wisc.edu

## A  Derivative of (6) w.r.t. $\boldsymbol{\beta}$

Give a censored list $\mathbf{a}$, define a mapping $\sigma$ that maps a state to its position in $\mathbf{a}$; that is, $\sigma(a_i) = i$. Let $\mathbf{N}^{(k)} = (\mathbf{I} - \mathbf{Q}^{(k)})^{-1}$. Hereafter, we drop the superscript $(k)$ from $\mathbf{Q}$, $\mathbf{R}$ and $\mathbf{N}$ when it's clear from the context.

Using $\partial(A^{-1})_{k\ell}/\partial A_{ij} = -(A^{-1})_{ki}(A^{-1})_{j\ell}$, the following identity becomes useful:

$$\begin{aligned}
\frac{\partial N_{k\ell}}{\partial Q_{ij}} &= \frac{\partial((\mathbf{I} - \mathbf{Q})^{-1})_{k\ell}}{\partial Q_{ij}} \\
&= \sum_{c,d} \frac{\partial((\mathbf{I} - \mathbf{Q})^{-1})_{k\ell}}{\partial(\mathbf{I} - \mathbf{Q})_{cd}} \frac{\partial(\mathbf{I} - \mathbf{Q})_{cd}}{\partial Q_{ij}} \\
&= \sum_{c,d} ((\mathbf{I} - \mathbf{Q})^{-1})_{kc}((\mathbf{I} - \mathbf{Q})^{-1})_{d\ell}\mathbb{1}_{\{c=i,d=j\}} \\
&= ((\mathbf{I} - \mathbf{Q})^{-1})_{ki}((\mathbf{I} - \mathbf{Q})^{-1})_{j\ell} \\
&= N_{ki}N_{jl}.
\end{aligned}$$

The derivative of $\mathbf{P}$ w.r.t. $\boldsymbol{\beta}$ is given as follows:

$$\begin{aligned}
\frac{\partial P_{rc}}{\partial \beta_{ij}} &= \mathbb{1}\{r = i\} \left( \frac{\mathbb{1}\{j = c\}e^{\beta_{ic}}(\sum_{\ell=1}^{n} e^{\beta_{i\ell}}) - e^{\beta_{ic}}e^{\beta_{ij}}}{(\sum_{\ell=1}^{n} e^{\beta_{i\ell}})^2} \right) \\
&= \mathbb{1}\{r = i\}(-P_{ic}P_{ij} + \mathbb{1}\{j = c\}P_{ic}).
\end{aligned}$$

The derivative of $\log \mathbb{P}(a_{k+1} \mid a_{1:k})$ with respect to $\boldsymbol{\beta}$ is

$$\begin{aligned}
\frac{\partial \log \mathbb{P}(a_{k+1} \mid a_{1:k})}{\partial \beta_{ij}} &= \mathbb{P}(a_{k+1} \mid a_{1:k})^{-1} \sum_{\ell=1}^{k} \frac{\partial(N_{k\ell}R_{\ell 1})}{\partial \beta_{ij}} \\
&= \mathbb{P}(a_{k+1} \mid a_{1:k})^{-1} \left( \sum_{\ell=1}^{k} \frac{\partial N_{k\ell}}{\partial \beta_{ij}} R_{\ell 1} + N_{k\ell} \frac{\partial R_{\ell 1}}{\partial \beta_{ij}} \right)
\end{aligned}$$

We need to compute $\frac{\partial N_{k\ell}}{\partial \beta_{ij}}$:

$$\frac{\partial N_{k\ell}}{\partial \beta_{ij}} = \sum_{c,d=1}^{k} \frac{\partial((\mathbf{I}-\mathbf{Q})^{-1})_{k\ell}}{\partial(\mathbf{I}-\mathbf{Q})_{cd}} \frac{\partial(\mathbf{I}-\mathbf{Q})_{cd}}{\partial \beta_{ij}}$$

$$= \sum_{c,d=1}^{k} (-1)N_{kc}N_{d\ell} \cdot (-1)\mathbb{1}_{\{a_c=i\}}(-P_{ia_d}P_{ij} + \mathbb{1}_{\{a_d=j\}}P_{ia_d})$$

$$= \mathbb{1}_{\{\sigma(i)\le k\}} N_{k\sigma(i)} \sum_{d=1}^{k} N_{d\ell}(-P_{ia_d}P_{ij} + \mathbb{1}_{\{a_d=j\}}P_{ia_d}),$$

where $\sigma(i) \le k$ means item $i$ appeared among the first $k$ items in the censored list $\mathbf{a}$.

Then,

$$\sum_{\ell=1}^{k} \frac{\partial N_{k\ell}}{\partial \beta_{ij}} R_{\ell 1} = \mathbb{1}_{\{\sigma(i)\le k\}} N_{k\sigma(i)} \sum_{\ell,d=1}^{k} N_{d\ell}(-P_{ia_d}P_{ij} + \mathbb{1}_{\{a_d=j\}}P_{ia_d})R_{\ell 1}$$

$$= \mathbb{1}_{\{\sigma(i)\le k\}} N_{k\sigma(i)} \left( -P_{ij}\sum_{d=1}^{k} P_{ia_d}\sum_{\ell=1}^{k} N_{d\ell}R_{\ell 1} + \sum_{d=1}^{k}\mathbb{1}_{\{a_d=j\}}P_{ia_d}\sum_{\ell=1}^{k} N_{d\ell}R_{\ell 1}\right)$$

$$= \mathbb{1}_{\{\sigma(i)\le k\}} N_{k\sigma(i)} \left( -P_{ij}(\mathbf{QNR})_{\sigma(i)1} + \mathbb{1}_{\{\sigma(j)\le k\}}P_{ij}(\mathbf{NR})_{\sigma(j)1}\right)$$

and

$$\sum_{\ell=1}^{k} N_{k\ell}\frac{\partial R_{\ell 1}}{\partial \beta_{ij}} = \sum_{\ell=1}^{k} N_{kl}\mathbb{1}_{\{\ell=\sigma(i)\}}\left(-P_{ia_{k+1}}P_{ij} + \mathbb{1}_{\{a_{k+1}=j\}}P_{ia_{k+1}}\right)$$

$$= \mathbb{1}_{\{\sigma(i)\le k\}} N_{k\sigma(i)}\left(-P_{ia_{k+1}}P_{ij} + \mathbb{1}_{\{a_{k+1}=j\}}P_{ia_{k+1}}\right).$$

Putting everything together,

$$\frac{\partial \log \mathbb{P}(a_{k+1} \mid a_{1:k})}{\partial \beta_{ij}}$$

$$= \frac{\mathbb{1}_{\{\sigma(i)\le k\}} N_{k\sigma(i)}}{\mathbb{P}(a_{k+1}\mid a_{1:k})}(-P_{ij}(\mathbf{QNR})_{\sigma(i)1} + \mathbb{1}_{\{\sigma(j)\le k\}}P_{ij}(\mathbf{NR})_{\sigma(j)1}$$

$$- P_{ia_{k+1}}P_{ij} + \mathbb{1}_{\{a_{k+1}=j\}}P_{ia_{k+1}})$$

$$= \frac{\mathbb{1}_{\{\sigma(i)\le k\}} N_{k\sigma(i)}P_{ij}}{\mathbb{P}(a_{k+1}\mid a_{1:k})}\left(-(\mathbf{QNR})_{\sigma(i)1} + \mathbb{1}_{\{\sigma(j)\le k\}}(\mathbf{NR})_{\sigma(j)1}\ P_{ia_{k+1}}\left(\frac{\mathbb{1}_{\{a_{k+1}=j\}}}{P_{ij}}-1\right)\right)$$

for all $i \ne j$.

## B   The Proof of Theorem 2

We first claim that *(i)* there must be a recurrent state $i$ in a censored list where $i \in W_k$ for some $k$. Then, it suffices to show that given *(i)* is true, *(ii)* recurrent states outside $W_k$ cannot appear, *(iii)* every states in $W_k$ must appear, and *(iv)* a transient state cannot appear after a recurrent state.

*(i)*: suppose there is no recurrent state in a censored list $\mathbf{a} = (a_{1:M})$. Then, every state $a_i$, $i \in [M]$, is a transient state. Since the underlying random walk runs indefinitely in finite state space, there must be a state $a_j$, $j \in [M]$, that is visited infinitely many times. This contradicts the fact that $a_j$ is a transient state.

Suppose a recurrent state $i \in W_k$ was visited. Then,

*(ii)*: the random walk cannot escape $W_k$ since $W_k$ is closed.

*(iii)*: the random walk will reach to every state in $W_k$ in finite time since $W_k$ is finite and irreducible.

*(iv)*: the same reason as *(iii)*.

## C  The Proof of Theorem 4

It suffices to show that $\mathbb{P}(a_{k+1} \mid a_{1:k}; \mathbf{P}) = \mathbb{P}(a_{k+1} \mid a_{1:k}; \mathbf{P}')$, where $\mathbf{a} = (a_1, \ldots, a_M)$ and $k \leq M - 1$. Define submatrices $(\mathbf{Q}, \mathbf{R})$ and $(\mathbf{Q}', \mathbf{R}')$ from $\mathbf{P}$ and $\mathbf{P}'$, respectively, as in (2). Note that $\mathbf{Q}' = \text{diag}(q_{1:k}) + (\mathbf{I} - \text{diag}(q_{1:k}))\mathbf{Q}$ and $\mathbf{R}' = (\mathbf{I} - \text{diag}(q_{1:k}))\mathbf{R}$.

$$
\begin{aligned}
\mathbb{P}(a_{k+1} \mid a_{1:k}; \mathbf{P}') &= \left(\mathbf{I} - \text{diag}(q_{1:k}) - (\mathbf{I} - \text{diag}(q_{1:k}))\mathbf{Q}\right)^{-1} (\mathbf{I} - \text{diag}(q_{1:k}))\mathbf{R} \\
&= (\mathbf{I} - \mathbf{Q})^{-1}(\mathbf{I} - \text{diag}(q_{1:k}))^{-1}(\mathbf{I} - \text{diag}(q_{1:k}))\mathbf{R} \\
&= \mathbb{P}(a_{k+1} \mid a_{1:k}; \mathbf{P})
\end{aligned}
$$

## D  The Proof of Theorem 5

Suppose $(\boldsymbol{\pi}, \mathbf{P}) \neq (\boldsymbol{\pi}', \mathbf{P}')$. We show that there exists a censored list $\mathbf{a}$ such that $\mathbb{P}(\mathbf{a}; \boldsymbol{\pi}, \mathbf{P}) \neq \mathbb{P}(\mathbf{a}; \boldsymbol{\pi}', \mathbf{P}')$.

**Case 1**: $\boldsymbol{\pi} \neq \boldsymbol{\pi}'$.

It follows that $\pi_i \neq \pi_i'$ for some $i$. Note that the marginal probability of observing $i$ as the first item in a censored list is $\sum_{\mathbf{a} \in \mathcal{D}: a_1 = i} \mathbb{P}(\mathbf{a}; \boldsymbol{\pi}, \mathbf{P}) = \pi_i$. Then,

$$
\sum_{\mathbf{a} \in \mathcal{D}: a_1 = i} \mathbb{P}(\mathbf{a}; \boldsymbol{\pi}, \mathbf{P}) = \pi_i \neq \pi_i' = \sum_{\mathbf{a} \in \mathcal{D}: a_1 = i} \mathbb{P}(\mathbf{a}; \boldsymbol{\pi}', \mathbf{P}').
$$

which implies that there exists a censored list $\mathbf{a}$ for which $\mathbb{P}(\mathbf{a}; \pi, \mathbf{P}) \neq \mathbb{P}(\mathbf{a}; \pi', \mathbf{P}')$.

**Case 2**: $\boldsymbol{\pi} = \boldsymbol{\pi}'$ but $\mathbf{P} \neq \mathbf{P}'$.

It follows that $P_{ij} \neq P_{ij}'$ for some $i$ and $j$. Then, we compute the marginal probability of observing $(i, j)$ as the first two items in a censored list, which results in

$$
\sum_{\mathbf{a} \in \mathcal{D}: \substack{a_1 = i, \\ a_2 = j}} \mathbb{P}(\mathbf{a}; \boldsymbol{\pi}, \mathbf{P}) = \pi_i P_{ij} \neq \pi_i' P_{ij}' = \sum_{\mathbf{a} \in \mathcal{D}: \substack{a_1 = i, \\ a_2 = j}} \mathbb{P}(\mathbf{a}; \boldsymbol{\pi}', \mathbf{P}').
$$

Then, there exists a censored list $\mathbf{a}$ for which $\mathbb{P}(\mathbf{a}; \pi, \mathbf{P}) \neq \mathbb{P}(\mathbf{a}; \pi', \mathbf{P}')$.

## E  Results Required for Theorem 6

Throughout, assume $\boldsymbol{\theta} = (\boldsymbol{\pi}^\top, \mathbf{P}_{1\cdot}, \ldots, \mathbf{P}_{n\cdot})^\top$. Let $\text{supp}(\boldsymbol{\theta})$ be the set of nonzero dimensions of $\boldsymbol{\theta}$: $\text{supp}(\boldsymbol{\theta}) = \{i \mid \theta_i > 0\}$. Lemma 1 shows conditions on which $\mathcal{Q}^*(\boldsymbol{\theta})$ and $\widehat{\mathcal{Q}}_m(\boldsymbol{\theta})$ are above $-\infty$.

**Lemma 1.** *Assume A1. Then,*

$$
supp(\boldsymbol{\theta}) \supseteq supp(\boldsymbol{\theta}^*) \iff \mathcal{Q}^*(\boldsymbol{\theta}) > -\infty \tag{7}
$$

$$
supp(\boldsymbol{\theta}) \supseteq supp(\boldsymbol{\theta}^*) \implies \widehat{\mathcal{Q}}_m(\boldsymbol{\theta}) > -\infty, \forall m. \tag{8}
$$

*Proof.* Define two vectors of probabilities w.r.t. $\boldsymbol{\theta}$ and $\boldsymbol{\theta}^*$: $\mathbf{q} = [q_\mathbf{a} = \mathbb{P}(\mathbf{a}; \boldsymbol{\theta})]_{\mathbf{a} \in \mathcal{D}}$ and $\mathbf{q}^* = [q_\mathbf{a}^* = \mathbb{P}(\mathbf{a}; \boldsymbol{\theta}^*)]_{\mathbf{a} \in \mathcal{D}}$. Note that

$$
\text{supp}(\mathbf{q}) \supseteq \text{supp}(\mathbf{q}^*) \iff \mathcal{Q}^*(\boldsymbol{\theta}) > -\infty
$$

by the definition of $\mathcal{Q}^*(\boldsymbol{\theta})$. Thus, for (7), it suffices to show that

$$
\text{supp}(\boldsymbol{\theta}) \supseteq \text{supp}(\boldsymbol{\theta}^*) \iff \text{supp}(\mathbf{q}) \supseteq \text{supp}(\mathbf{q}^*).
$$

( $\implies$ ) The LHS implies that the directed graph enduced by $\boldsymbol{\theta}$ includes the graph enduced by $\boldsymbol{\theta}^*$; a path that is possible w.r.t. $\boldsymbol{\theta}^*$ is also possible w.r.t. $\boldsymbol{\theta}$. Recall that a list is generated by a random walk. Let $\mathbf{a} \in \text{supp}(\mathbf{q}^*)$. There exists a random walk under $\boldsymbol{\theta}^*$ that generates $\mathbf{a}$. Then, the same random walk is also possible under $\boldsymbol{\theta}$, which implies $\mathbf{a} \in \text{supp}(\mathbf{q})$.

( $\impliedby$ ) Suppose the LHS is false. Then, there exists $(i, j)$ s.t. $P_{ij} = 0$ and $P_{ij}^* > 0$. Consider a list $\mathbf{a}$ such that it has nonzero probability w.r.t. $\boldsymbol{\theta}^*$ (that is, $q_\mathbf{a}^* > 0$), and its first two items are $i$ then $j$. Since $P_{ij} = 0$, $q_\mathbf{a} = 0$. However, the RHS implies that $q_\mathbf{a} > 0$ since $q_\mathbf{a}^* > 0$: a contradiction.

For (8),

$$\mathrm{supp}(\boldsymbol{\theta}) \supseteq \mathrm{supp}(\boldsymbol{\theta}^*) \implies \mathcal{Q}^*(\boldsymbol{\theta}) > -\infty \implies \widehat{\mathcal{Q}}_m(\boldsymbol{\theta}) > -\infty, \forall m,$$

where the last implication is due to the fact that a censored list $\mathbf{a}^{(i)}$ that appears in $\widehat{\mathcal{Q}}_m(\boldsymbol{\theta})$ is generated by $\boldsymbol{\theta}^*$, so the term $\log \mathbb{P}(\mathbf{a}^{(i)}; \boldsymbol{\theta})$ also appears in $\mathcal{Q}^*(\boldsymbol{\theta})$.

$\square$

**Lemma 2.** *Assume A1. Then, $\boldsymbol{\theta}^*$ is the unique maximizer of $\mathcal{Q}^*(\boldsymbol{\theta})$.*

*Proof.* If $\boldsymbol{\theta}$ satisfies $\mathrm{supp}(\boldsymbol{\theta}) \not\supseteq \mathrm{supp}(\boldsymbol{\theta}^*)$, then $\mathcal{Q}^*(\boldsymbol{\theta}) = -\infty$ by Lemma 1, so such $\boldsymbol{\theta}$ cannot be a maximizer. Thus, it is safe to restrict our attention to $\boldsymbol{\theta}$'s whose support include that of $\boldsymbol{\theta}^*$: $\mathrm{supp}(\boldsymbol{\theta}) \supseteq \mathrm{supp}(\boldsymbol{\theta}^*)$.

Recall the definition of $\mathcal{Q}^*(\boldsymbol{\theta})$:

$$\mathcal{Q}^*(\boldsymbol{\theta}) = \sum_{\mathbf{a} \in \mathcal{D}} \mathbb{P}(\mathbf{a}; \boldsymbol{\theta}^*) \log \mathbb{P}(\mathbf{a}; \boldsymbol{\theta}) \propto -\mathrm{KL}(\boldsymbol{\theta}^* \| \boldsymbol{\theta}),$$

where $\mathrm{KL}(\boldsymbol{\theta}^* \| \boldsymbol{\theta})$ is well defined since $\mathrm{supp}(\boldsymbol{\theta}) \supseteq \mathrm{supp}(\boldsymbol{\theta}^*)$. Due to the identifiability of the model (Theorem 5) and the unique minimizer property of the KL-divergence, $\boldsymbol{\theta}^*$ is the unique maximizer.

$\square$

We denote by $\mathrm{decomp}(\boldsymbol{\theta}) = \{T, W_1, \dots, W_K\}$ the decomposition enduced by $\boldsymbol{\theta}$ as in Theorem 1.

**Lemma 3.** $\mathrm{supp}(\widehat{\boldsymbol{\theta}}_m) \supseteq \mathrm{supp}(\boldsymbol{\theta}^*)$ *for large enough $m$. Furthermore, $\mathrm{decomp}(\widehat{\boldsymbol{\theta}}_m) = \mathrm{decomp}(\boldsymbol{\theta}^*)$ for large enough $m$.*

*Proof.* Note that due to the strong law of large numbers, a list $\mathbf{a}$ is valid in the true model $\boldsymbol{\theta}^*$ must appear in $D_m$ for large enough $m$. Since the number of censored lists that can be generated by $\boldsymbol{\theta}^*$ is finite, one observes every valid censored list in the true model $\boldsymbol{\theta}^*$; that is, there exists $m'$ such that

$$m \geq m' \implies \{\mathbf{a} \mid \mathbf{a} \in D_m\} = \{\mathbf{a} \mid \mathbb{P}(\mathbf{a}; \boldsymbol{\theta}^*) > 0\}.$$

For the first statement, assume that $m \geq m'$. Since we observe every valid list in $\boldsymbol{\theta}^*$, by the definition of $\widehat{\mathcal{Q}}_m(\boldsymbol{\theta})$, the following holds true:

$$\forall \boldsymbol{\theta} \in \boldsymbol{\Theta}, \ \widehat{\mathcal{Q}}_m(\boldsymbol{\theta}) > -\infty \iff \mathcal{Q}^*(\boldsymbol{\theta}) > -\infty.$$

Then, using Lemma 1,

$$\widehat{\mathcal{Q}}_m(\widehat{\boldsymbol{\theta}}_m) > -\infty \implies \mathcal{Q}^*(\widehat{\boldsymbol{\theta}}_m) > -\infty \implies \mathrm{supp}(\widehat{\boldsymbol{\theta}}_m) \supseteq \mathrm{supp}(\boldsymbol{\theta}^*).$$

For the second statement, assume $m \geq m'$. Let $\mathrm{decomp}(\widehat{\boldsymbol{\theta}}_m) = \{\widehat{T}, \widehat{W}_1, \dots, \widehat{W}_{\widehat{K}}\}$ and $\mathrm{decomp}(\boldsymbol{\theta}^*) = \{T^*, W_1^*, \dots, W_{K^*}^*\}$. Furthermore, define $\widehat{\tau}(i)$ to be the index of the closed irreducible set in $\mathrm{decomp}(\widehat{\boldsymbol{\theta}}_m)$ to which $i$ belongs, and define $\tau^*(i)$ similary.

Suppose that the data $D_m$ contains every valid list in $\boldsymbol{\theta}^*$, but $\mathrm{decomp}(\widehat{\boldsymbol{\theta}}_m) \neq \mathrm{decomp}(\boldsymbol{\theta}^*)$. There are four cases. In each case, we show that there exists a list that is valid in $\boldsymbol{\theta}^*$ but not in $\widehat{\boldsymbol{\theta}}_m$, which means that the log likelihood of $\widehat{\boldsymbol{\theta}}_m$ is $-\infty$. This is a contradiction in that $\widehat{\boldsymbol{\theta}}_m$ is the MLE.

**Case 1** : $\exists s_1$ s.t. $s_1$ is transient in $\widehat{\boldsymbol{\theta}}$ but recurrent in $\boldsymbol{\theta}^*$.

Let $W_k^*$ be the closed irreducible set to which $s_1$ belongs and $L = |W_k^*|$. Use $\boldsymbol{\theta}^*$ to start a random walk from $s_1$ and generate a censored list $\mathbf{a}$, which consists of all states in $W_k^*$: $\mathbf{a} = (s_1, s_2, \dots, s_L)$. If $\mathbf{a}$ is invalid in $\widehat{\boldsymbol{\theta}}_m$, we have a contradiction. If not, $s_L$ must be recurrent in $\widehat{\boldsymbol{\theta}}_m$ by Theorem 2. Use $\boldsymbol{\theta}^*$ to generate a censored list $\mathbf{a}'$ that starts from $s_L$. Then, $s_1$ must appear after $s_L$ in $\mathbf{a}'$. However, this is impossible in $\widehat{\boldsymbol{\theta}}_m$ since $s_1$ is transient and $s_L$ is recurrent: a contradiction.

**Case 2** : $\exists t$ s.t. is transient in $\boldsymbol{\theta}^*$ but recurrent in $\widehat{\boldsymbol{\theta}}_m$.

For brevity, assume that $t$ is the only transient state in $\boldsymbol{\theta}^*$; this can be easily relaxed. Use $\boldsymbol{\theta}^*$ to generate a censored list that starts with $t$, say $\mathbf{a} = (t, s_1, \ldots, s_L)$. By Theorem 2, $\{s_{1:L}\}$ is a closed irreducible set in $\boldsymbol{\theta}^*$. Define $\mathbf{a}' = (s_{1:L})$, which is also valid in $\boldsymbol{\theta}^*$. Now, $\mathbf{a}$ may or may not be valid in $\widehat{\boldsymbol{\theta}}_m$. Assume that $\mathbf{a}$ is valid in $\widehat{\boldsymbol{\theta}}_m$ since otherwise we have a contradiction. Then, in $\widehat{\boldsymbol{\theta}}_m$, $\{t, s_{1:L}\}$ must be a closed irreducible set since $t$ is recurrent. Then, $\mathbf{a}' = (s_{1:L})$ is invalid in $\widehat{\boldsymbol{\theta}}_m$ since $t$ must be visited as well: a contradiction.

**Case 3.** $\exists(i, j)$ s.t. $\widehat{\tau}(i) = \widehat{\tau}(j)$ , but $\tau^*(i) \neq \tau^*(j)$.

Start a random walk from the state $i$ w.r.t. $\boldsymbol{\theta}^*$ and generate a censored list $\mathbf{a}$. By Theorem 2, the censored list $\mathbf{a}$ does not contain $j$. In $\widehat{\boldsymbol{\theta}}_m$, however, a censored list starting from $i$ must also output $j$ since $i$ and $j$ are in the same closed irreducible set. Thus, $\mathbf{a}$ is invalid in $\widehat{\boldsymbol{\theta}}_m$: a contradiction.

**Case 4.** $\exists(i, j)$ s.t. $\tau^*(i) = \tau^*(j)$ , but $\widehat{\tau}(i) \neq \widehat{\tau}(j)$.

Start a random walk from the state $i$ w.r.t. $\boldsymbol{\theta}^*$ and generate a censored list $\mathbf{a}$. By Theorem 2, the censored list $\mathbf{a}'$ must also contain $j$. In $\widehat{\boldsymbol{\theta}}_m$, however, a censored list starting from $i$ cannot output $j$ since $j$ is in a different closed irreducible set. Thus, $\mathbf{a}'$ is invalid in $\widehat{\boldsymbol{\theta}}_m$: a contradiction.

$\square$

**Lemma 4.** *Assume A1. Let $\{\widehat{\boldsymbol{\theta}}_{m_j}\}$ be a convergent subsequence of $\{\widehat{\boldsymbol{\theta}}_m\}$ and $\boldsymbol{\theta}'$ be its limit point: $\boldsymbol{\theta}' = \lim_{j\to\infty} \widehat{\boldsymbol{\theta}}_{m_j}$. Then, $\lim_{j\to\infty} \mathbb{P}(\mathbf{a}; \widehat{\boldsymbol{\theta}}_{m_j}) = \mathbb{P}(\mathbf{a}; \boldsymbol{\theta}')$ for all $\mathbf{a}$ that is valid in $\boldsymbol{\theta}^*$.*

*Proof.* There are exactly two case-by-case operators which causes the likelihood function to be discontinuous. The operators appear in (3) and (1), which respectively rely on the following conditions w.r.t. a list $\mathbf{a} = (a_{1:M})$:

$$(\mathbf{I} - \mathbf{Q}^{(k)})^{-1} \text{ exists, } \forall k \in [M - 1] \tag{9}$$
$$\mathbb{P}(s \mid a_{1:M}; \boldsymbol{\theta}) = 0, \forall s \in S \setminus \{a_{1:M}\}. \tag{10}$$

**Step 1**: claim that $\forall \boldsymbol{\theta} \in \boldsymbol{\Theta}$,

$$\text{supp}(\boldsymbol{\theta}) \supseteq \text{supp}(\boldsymbol{\theta}^*) \text{ and } \text{decomp}(\boldsymbol{\theta}) = \text{decomp}(\boldsymbol{\theta}^*) \implies \forall \mathbf{a} \text{ valid in } \boldsymbol{\theta}^* \text{ (9) and (10)}$$

To show (9), suppose it is false for some $k \in [M - 1]$ and some censored list $\mathbf{a} = (a_{1:M})$ valid in $\boldsymbol{\theta}^*$. The nonexistence of $(\mathbf{I} - \mathbf{Q}^{(k)})^{-1}$ implies that there is no path from $a_k$ to a state that is outside of $\{a_{1:k}\}$ whereas there is such a path w.r.t. $\boldsymbol{\theta}^*$. This contradicts $\text{supp}(\boldsymbol{\theta}) \supseteq \text{supp}(\boldsymbol{\theta}^*)$.

To show (10), consider a censored list $\mathbf{a} = (a_{1:M})$ that is valid in $\boldsymbol{\theta}^*$. By Theorem 2, the last state $a_M$ must be a recurrent state in a closed irreducible set $W$ w.r.t. $\boldsymbol{\theta}^*$. Since $\boldsymbol{\theta}$ has the same decomposition as $\boldsymbol{\theta}^*$ and every state in $W$ must be present in $\mathbf{a}$, no other state can appear after $a_M$. This implies (10).

Define

$$\boldsymbol{\Theta}' = \{\boldsymbol{\theta} \in \boldsymbol{\Theta} \mid \|\boldsymbol{\theta} - \boldsymbol{\theta}'\|_\infty < \min_i \theta_i', \text{decomp}(\boldsymbol{\theta}) = \text{decomp}(\boldsymbol{\theta}')\}.$$

**Step 2**: claim that $\mathbb{P}(\mathbf{a}; \boldsymbol{\theta})$ is a continuous function of $\boldsymbol{\theta}$ in the subspace $\boldsymbol{\Theta}'$, $\forall \mathbf{a}$ valid in $\boldsymbol{\theta}^*$.

Note that $\forall \boldsymbol{\theta} \in \boldsymbol{\Theta}'$,

$$
\begin{array}{ccccc}
\text{supp}(\boldsymbol{\theta}) & \supseteq & \text{supp}(\boldsymbol{\theta}') & \supseteq & \text{supp}(\boldsymbol{\theta}^*) \\
\text{decomp}(\boldsymbol{\theta}) & = & \text{decomp}(\boldsymbol{\theta}') & = & \text{decomp}(\boldsymbol{\theta}^*),
\end{array}
$$

where the first subset relation is due to the $\infty$-norm in the definition of $\boldsymbol{\Theta}'$, the second subset relation and the last equality is due to Lemma 3 and $\boldsymbol{\theta}' = \lim_{j\to\infty} \widehat{\boldsymbol{\theta}}_{m_j}$.

This implies, together with step 1, that $\forall \boldsymbol{\theta} \in \boldsymbol{\Theta}'$, (9) and (10) are satisfied, which effectively gets rid of the case-by-case operators in $\boldsymbol{\Theta}'$. This concludes the claim.

**Step 3**: $\lim_{j\to\infty} \mathbb{P}(\mathbf{a}; \widehat{\boldsymbol{\theta}}_{m_j}) = \mathbb{P}(\mathbf{a}; \boldsymbol{\theta}')$ for all $\mathbf{a}$ that is valid in $\boldsymbol{\theta}^*$.

Since $\widehat{\boldsymbol{\theta}}_{m_j} \to \boldsymbol{\theta}'$, there exists $J$ such that

$$j \geq J \implies ||\widehat{\boldsymbol{\theta}}_{m_j} - \boldsymbol{\theta}'||_\infty < \min_i \theta_i' .$$

Thus, after $J$, the sequence enters the subspace $\boldsymbol{\Theta}'$ in which $\mathbb{P}(\mathbf{a}; \boldsymbol{\theta})$ is continuous $\forall \mathbf{a}$ valid in $\boldsymbol{\theta}^*$, which concludes the claim.

$\square$

**Lemma 5.** *Assume A1. Let $\{\widehat{\boldsymbol{\theta}}_{m_j}\}$ be a convergent subsequence of $\{\widehat{\boldsymbol{\theta}}_m\}$ and $\boldsymbol{\theta}'$ be its limit point:* $\boldsymbol{\theta}' = \lim_{j\to\infty} \widehat{\boldsymbol{\theta}}_{m_j}$. *Then, $\mathcal{Q}^*(\boldsymbol{\theta}') > -\infty$.*

*Proof.* Suppose not: $\mathcal{Q}^*(\boldsymbol{\theta}') = -\infty$. Then, there exists a list $\mathbf{a}'$ that is valid in $\boldsymbol{\theta}^*$ whose likelihood w.r.t. $\boldsymbol{\theta}'$ converges to 0:
$$\exists \mathbf{a}' \text{ s.t. } \mathbb{P}(\mathbf{a}'; \boldsymbol{\theta}^*) > 0 \text{ and } \mathbb{P}(\mathbf{a}'; \boldsymbol{\theta}') = 0,$$
By Lemma 4, $\mathbb{P}(\mathbf{a}'; \boldsymbol{\theta}') = 0$ implies that $\lim_{j\to\infty} \mathbb{P}(\mathbf{a}'; \widehat{\boldsymbol{\theta}}_{m_j}) = 0$.

Let $0 < \epsilon < \mathbb{P}(\mathbf{a}'; \boldsymbol{\theta}^*)$. Denote by $\#\{\mathbf{a}'\}$ the number of occurrences of the list $\mathbf{a}'$ in $\{\mathbf{a}^{(1)}, \ldots, \mathbf{a}^{(m_j)}\}$. Then, the following statements hold:

$$\exists J_1 \text{ s.t. } j > J_1 \implies \left| \frac{\#\{\mathbf{a}'\}}{m_j} - \mathbb{P}(\mathbf{a}'; \boldsymbol{\theta}^*) \right| < \epsilon \tag{11}$$

$$\exists J_2 \text{ s.t. } j < J_2 \implies \left| \widehat{\mathcal{Q}}_{m_j}(\boldsymbol{\theta}^*) - \mathcal{Q}^*(\boldsymbol{\theta}^*) \right| < \epsilon \tag{12}$$

$$\exists J_3 \text{ s.t. } j > J_3 \implies \log \mathbb{P}(\mathbf{a}'; \widehat{\boldsymbol{\theta}}_{m_j}) < \frac{\mathcal{Q}^*(\boldsymbol{\theta}^*) - \epsilon}{\mathbb{P}(\mathbf{a}'; \boldsymbol{\theta}^*) - \epsilon}. \tag{13}$$

The first two statements are due to the law of large numbers, and the last statement is due to the convergence of $\mathbb{P}(\mathbf{a}'; \widehat{\boldsymbol{\theta}}_{m_j})$ to 0. Note that $\widehat{\mathcal{Q}}_{m_j}(\boldsymbol{\theta}^*) \leq \widehat{\mathcal{Q}}_{m_j}(\widehat{\boldsymbol{\theta}}_{m_j})$ since $\widehat{\boldsymbol{\theta}}_{m_j}$ is the maximizer of the function $\widehat{\mathcal{Q}}_{m_j}(\boldsymbol{\theta})$. Then, if $j > \max\{J_1, J_2, J_3\}$,

$$\begin{aligned}
\mathcal{Q}^*(\boldsymbol{\theta}^*) - \epsilon &\leq \widehat{\mathcal{Q}}_{m_j}(\boldsymbol{\theta}^*) \\
&\leq \widehat{\mathcal{Q}}_{m_j}(\widehat{\boldsymbol{\theta}}_{m_j}) \\
&= \left( \sum_{\mathbf{a}\neq\mathbf{a}'} \frac{\#\{\mathbf{a}\}}{m_j} \log \mathbb{P}(\mathbf{a}; \widehat{\boldsymbol{\theta}}_{m_j}) \right) + \frac{\#\{\mathbf{a}'\}}{m_j} \log \mathbb{P}(\mathbf{a}'; \widehat{\boldsymbol{\theta}}_{m_j}) \\
&< (\mathbb{P}(\mathbf{a}'; \boldsymbol{\theta}^*) - \epsilon) \log \mathbb{P}(\mathbf{a}'; \widehat{\boldsymbol{\theta}}_{m_j}) \\
&< \mathcal{Q}^*(\boldsymbol{\theta}^*) - \epsilon,
\end{aligned}$$

where the last inequality is due to (13). This is a contradiction. $\square$

**Lemma 6.** *Assume A1. Let $\{\widehat{\boldsymbol{\theta}}_{m_j}\}$ be a convergent subsequence of $\{\widehat{\boldsymbol{\theta}}_m\}$. Let $\boldsymbol{\theta}' = \lim_{j\to\infty} \widehat{\boldsymbol{\theta}}_{m_j}$. Then, $\lim_{j\to\infty} \widehat{\mathcal{Q}}_{m_j}(\widehat{\boldsymbol{\theta}}_{m_j}) = \mathcal{Q}^*(\boldsymbol{\theta}')$.*

*Proof.* The idea is that we can have a compact ball around the limit point $\boldsymbol{\theta}'$ and show that the log likelihood $\widehat{\mathcal{Q}}_{m_j}(\boldsymbol{\theta})$ converges uniformly on the ball. Then, after the sequence $\widehat{\boldsymbol{\theta}}_{m_j}$ gets in the ball, we can use the uniform convergence of the log likelihood.

Let $B_{\boldsymbol{\theta}'}(r) = \{\boldsymbol{\theta} \in \boldsymbol{\Theta} \mid ||\boldsymbol{\theta} - \boldsymbol{\theta}'||_\infty \leq r\}$ be an $\infty$-norm ball around $\boldsymbol{\theta}'$. Choose $\epsilon' < \min_{i\in\text{supp}(\boldsymbol{\theta})} \theta_i$. We claim that

$$\forall \boldsymbol{\theta} \in B_{\boldsymbol{\theta}'}(\epsilon'), \ \mathcal{Q}^*(\boldsymbol{\theta}) > -\infty \text{ and } \widehat{\mathcal{Q}}_m(\boldsymbol{\theta}) > -\infty, \forall m. \tag{14}$$

Let $\boldsymbol{\theta} \in B_{\boldsymbol{\theta}'}(\epsilon')$. By the definition of the ball $B_{\boldsymbol{\theta}'}(\epsilon')$, $\text{supp}(\boldsymbol{\theta}) \supseteq \text{supp}(\boldsymbol{\theta}')$. Note that $\mathcal{Q}^*(\boldsymbol{\theta}') > -\infty$ by Lemma 5. By Lemma 1, $\text{supp}(\boldsymbol{\theta}') \supseteq \text{supp}(\boldsymbol{\theta}^*)$:

$$\text{supp}(\boldsymbol{\theta}) \supseteq \text{supp}(\boldsymbol{\theta}') \supseteq \text{supp}(\boldsymbol{\theta}^*).$$

This then, again by Lemma 1, implies the claim. Now, $\widehat{\mathcal{Q}}_{m_j}(\boldsymbol{\theta})$ converges to $\mathcal{Q}^*(\boldsymbol{\theta})$ uniformly on the ball $B_{\boldsymbol{\theta}'}(\epsilon')$ since the function is continuous on the ball that is compact.

Let $0 < \epsilon < 2\epsilon'$. Note

$$P\left(\|\widehat{\boldsymbol{\theta}}_{m_j} - \boldsymbol{\theta}'\|_\infty > \epsilon/2\right) \to 0 \tag{15}$$

$$P\left(\sup_{\boldsymbol{\theta} \in B_{\boldsymbol{\theta}'}(\epsilon')} \left|\widehat{\mathcal{Q}}_{m_j}(\boldsymbol{\theta}) - \mathcal{Q}^*(\boldsymbol{\theta})\right| > \epsilon/2\right) \to 0 \tag{16}$$

$$P\left(|\mathcal{Q}^*(\widehat{\boldsymbol{\theta}}_{m_j}) - \mathcal{Q}^*(\boldsymbol{\theta}')| > \epsilon/2\right) \to 0. \tag{17}$$

(15) is due to the convergence of $\{\widehat{\boldsymbol{\theta}}_{m_j}\}$. (16) holds because of the uniform convergence on the ball $B_{\boldsymbol{\theta}'}(\epsilon')$. (17) holds because $\mathcal{Q}^*(\boldsymbol{\theta})$ is continuous at $\boldsymbol{\theta}$.

Recall we want to show $\mathbb{P}(|\widehat{\mathcal{Q}}_{m_j}(\widehat{\boldsymbol{\theta}}_{m_j}) - \mathcal{Q}^*(\boldsymbol{\theta}')| > \epsilon) \to 0$. Note:

$$\mathbb{P}(|\widehat{\mathcal{Q}}_{m_j}(\widehat{\boldsymbol{\theta}}_{m_j}) - \mathcal{Q}^*(\boldsymbol{\theta}')| > \epsilon)$$
$$\leq \mathbb{P}(|\widehat{\mathcal{Q}}_{m_j}(\widehat{\boldsymbol{\theta}}_{m_j}) - \mathcal{Q}^*(\widehat{\boldsymbol{\theta}}_{m_j})| > \epsilon/2) + \mathbb{P}(|\mathcal{Q}^*(\widehat{\boldsymbol{\theta}}_{m_j}) - \mathcal{Q}^*(\boldsymbol{\theta}')| > \epsilon/2).$$

The second term goes to zero by (17). It remains to show that the first term goes to 0:

$$\mathbb{P}(|\widehat{\mathcal{Q}}_{m_j}(\widehat{\boldsymbol{\theta}}_{m_j}) - \mathcal{Q}^*(\widehat{\boldsymbol{\theta}}_{m_j})| > \epsilon/2)$$
$$\leq \mathbb{P}\left(|\widehat{\mathcal{Q}}_{m_j}(\widehat{\boldsymbol{\theta}}_{m_j}) - \mathcal{Q}^*(\widehat{\boldsymbol{\theta}}_{m_j})| > \epsilon/2 \,\Big|\, \|\widehat{\boldsymbol{\theta}}_{m_j} - \boldsymbol{\theta}'\|_\infty > \epsilon/2\right) \mathbb{P}\left(\|\widehat{\boldsymbol{\theta}}_{m_j} - \boldsymbol{\theta}'\|_\infty > \epsilon/2\right) +$$
$$\mathbb{P}\left(\left\{|\widehat{\mathcal{Q}}_{m_j}(\widehat{\boldsymbol{\theta}}_{m_j}) - \mathcal{Q}^*(\widehat{\boldsymbol{\theta}}_{m_j})| > \epsilon/2\right\} \cap \left\{\|\widehat{\boldsymbol{\theta}}_{m_j} - \boldsymbol{\theta}'\|_\infty \leq \epsilon/2\right\}\right).$$

The first term goes to zero by (15). The second term also goes to zero as follows, which completes the proof:

$$\mathbb{P}\left(\left\{|\widehat{\mathcal{Q}}_{m_j}(\widehat{\boldsymbol{\theta}}_{m_j}) - \mathcal{Q}^*(\widehat{\boldsymbol{\theta}}_{m_j})| > \epsilon/2\right\} \cap \left\{\|\widehat{\boldsymbol{\theta}}_{m_j} - \boldsymbol{\theta}'\|_\infty \leq \epsilon/2\right\}\right)$$
$$\leq \mathbb{P}\left(\left\{\sup_{\boldsymbol{\theta} \in B_{\boldsymbol{\theta}'}(\epsilon/2)} |\widehat{\mathcal{Q}}_{m_j}(\boldsymbol{\theta}) - \mathcal{Q}^*(\boldsymbol{\theta})| > \epsilon/2\right\} \cap \left\{\|\widehat{\boldsymbol{\theta}}_{m_j} - \boldsymbol{\theta}'\|_\infty \leq \epsilon/2\right\}\right)$$
$$\leq \mathbb{P}\left(\sup_{\boldsymbol{\theta} \in B_{\boldsymbol{\theta}'}(\epsilon/2)} |\widehat{\mathcal{Q}}_{m_j}(\boldsymbol{\theta}) - \mathcal{Q}^*(\boldsymbol{\theta})| > \epsilon/2\right)$$
$$\leq \mathbb{P}\left(\sup_{\boldsymbol{\theta} \in B_{\boldsymbol{\theta}'}(\epsilon')} |\widehat{\mathcal{Q}}_{m_j}(\boldsymbol{\theta}) - \mathcal{Q}^*(\boldsymbol{\theta})| > \epsilon/2\right) \to 0,$$

where the last line is due to (16). $\qquad\square$