[Reviews · NeurIPS 2015]

Submitted by Assigned_Reviewer_1

An initial-visit emitting random walk is a random walk that emits an output only when visiting a state for the first time. Previous researchers have used these walks to model human behavior data on memory search tasks (e.g., name as many animals as you can). The contributions of this paper are that it proposes a new and efficient way of computing a likelihood function for the model, it proves that the resulting maximum likelihood estimates are consistent, it presents a way of estimating parameter values using a regularized likelihood function, and it shows, via computer simulation, that the proposed parameter estimation procedure performs better than alternatives on a toy task and on real-world data collected (by others) in experiments with people.

I should start by stating that this paper relies on areas of mathematics that I know nothing about.

That said, I have a mixed opinion of this paper. On the one hand, the proposed way of computing a likelihood for the model seems extremely clever and a clear technical advance over previous efforts. In addition, the paper is very well-written. On the other hand, the paper is likely to appeal to a very small segment of the NIPS community. Initial-visit emitting random walks would seem to have limited applicability. Indeed, to my knowledge, they have only been studied by psychologists with interests in mathematical models of semantic memory search (a surprisingly small sub-community). In addition, the point of using these random walks is that they reveal something interesting about human memory search. However, although the current paper provides an improvement in fitting parameter values for these walks, it does not demonstrate any payoff in terms of revealing new information about human memory search. That is, the paper demonstrates improved parameter estimation but no payoff from these improved parameter estimates.

But, but, but...perhaps I'm being too harsh. Maybe someone else will be able to use the parameter estimation procedure proposed here to learn something new about human memory search. I hope so.
Summary: The paper provides a technical advance. However, this advance is a narrow contribution in the sense that the problem studied here is likely to be of interest to a very small community. Moreover, the payoff from this technical advance is not clear.

Submitted by Assigned_Reviewer_2

This paper is very well written.

It's easy to understand yet it does not sacrifice rigor.

The only typos I found were "both chains has two transient states" (change "has" to "have"), and "before exhausting every states the walk could visit" ("states" should be singular).

The reformulation of the state machine to allow for tractable likelihood computation is clever.

For the proof of theorem 5, the assumption is that there are no self transitions in the matrix.

I wonder what that means for the underlying process that's being modeled.

In concrete terms, it means that if I think "goat" I cannot think "goat" on the next time step, I have to think "chicken" (or anything other than "goat") and then return to thinking "goat".

It would be helpful to elaborate on the cognitive plausibility of that assumption, though I suppose the ultimate proof is in the empirical pudding.

In the toy experiments, why does every walk include every node in the

output?

That would seem to be the ideal case for the algorithms, but it is probably not realistic.

I like the experiments with the verbal fluency data.

The methodology seems to be sound.

It would be nice to see some discussion of the "absolute" quality of the results.

I don't have any great ideas on how to do that, but what I'm wondering is how much more room is there for improvement in the estimation procedure.

One concern I have is the breadth of appeal of the domain and method. I like the content, but it's not clear that the form of the underlying generative model has many other applications.
Summary: A good paper that would benefit from clarifying a few assumptions and expanding on possible other applications of the ideas.

Submitted by Assigned_Reviewer_3

Quality: I found the results both theoretical and experiment quite compelling. One small comment: it would have been useful to get some quantification of runtimes for the stochastic gradient descent vs batch RegML to get a better feeling for the advantage of doing things online.

Clarity: paper is well written. Originality: the results are new and the link to past work clearly delineated in the discussion. Significance: useful for modelling human behaviour in memory search tasks, and thus relevant for cogSci community.
Summary: The paper present a new algorithm for ML parameter estimation for INVITE (random walk on graph with observations listing only first visit of states). Nice theoretical results supported by experiments on artificial and human memory search data.

Author Feedback
Author rebuttal: To R2:
We agree with your comment that "no self-transition" assumption might not be plausible as a cognitive model. To clarify, we are not arguing that humans do not make self-transitions. Instead of fixing self-transitions to 0, we could have fixed them to 0.1 without hurting any results in the paper. The reason is that, as noted in Theorem 4, assigning any value to the self-transitions does not make a difference in the INVITE model. That said, if we observe how long it takes to generate each item in a list, we would be able to learn the self-transition probabilities by adding a timing model to INVITE. In fact, incorporating the timing information is one of our immediate future works.

The purpose of the toy experiment is to empirically verify the consistency of the INVITE estimator. When generating a list, we exhaust all the items because the consistency theorem assumes that the underlying, unobserved random walk of INVITE continues indefinitely. We could generate the toy data without exhausting all the items by introducing a length distribution. In that case, we believe that INVITE is still consistent since it is trivial to modify our consistency result to assume a length distribution on the censored list.

To R6:
That is a good point. The underlying transition matrix P is likely to be different for every person. At this point, we do not have a sufficient number of lists gathered from each person to reliably estimate P at the individual level. Instead, we have different age groups from which we can estimate P for each group separately to study the difference between groups. In fact, cognitive scientists are interested in how aging affects the human memory search ability, so this would be a nice future work.